# Social Determinants of Health and Distance Learning in Italy in the Era of the SARS-CoV-2 Pandemic

**DOI:** 10.3390/ijerph19095741

**Published:** 2022-05-09

**Authors:** Arianna Dondi, Jacopo Lenzi, Egidio Candela, Sugitha Sureshkumar, Francesca Morigi, Carlotta Biagi, Marcello Lanari

**Affiliations:** 1Paediatric Emergency Unit, IRCCS Azienda Ospedaliero-Universitaria di Bologna, 40138 Bologna, Italy; arianna.dondi@aosp.bo.it (A.D.); carlotta.biagi@aosp.bo.it (C.B.); marcello.lanari@unibo.it (M.L.); 2Department of Biomedical and Neuromotor Sciences, University of Bologna, 40125 Bologna, Italy; jacopo.lenzi2@unibo.it; 3Specialty School of Pediatrics, Alma Mater Studiorum, University of Bologna, 40126 Bologna, Italy; francesca.morigi@gmail.com; 4Institute of Global Health, University of Geneva, 1205 Geneva, Switzerland; sugitha.sureshkumar@etu.unige.ch

**Keywords:** social determinants of health, distance learning, remote learning, SARS-CoV-2 pandemic, COVID-19, social inequities

## Abstract

Objectives: To investigate the experiences by distance learning (DL) method during the first wave of the SARS-CoV-2 pandemic in Italy, and to search for correlations with purported experiences and respective levels of social determinants of health (SDH). Study design and methods: Cross-sectional online survey, investigating various SDH and parents’ attitude towards DL, proposed 6 months after the beginning of the pandemic to a sample population of parents with school-aged children throughout Italy. Results: A total of 3791 questionnaires were analyzed. Non-Italian parents complained more frequently of difficulties in providing support to their children in DL due to poor digital skills (*p* = 0.01), lack of good-quality digital equipment (*p* = 0.01), problems with the Italian language (*p* < 0.001), and a lower level of education (*p* < 0.001). When parents lived apart, greater difficulties in concentration in children using DL (*p* = 0.05) and a lower parental capacity to support DL (*p* = 0.002) were reported. Adequate digital structures appeared related to living in owned compared to rented property, higher levels of parental education, and better familial financial situations. Conclusions: Students from families with financial difficulties and low levels of parental education, or even those living in houses for rent or having separated parents, may be disadvantaged in an educational context since the introduction of DL.

## 1. Introduction

Social determinants of health (SDH) are the set of conditions in which people are born, grow, live, work, and age, that affect their state of health [1]. Overall, SDH include political systems, economic systems, development programs, and can be grouped mainly into six fields: economic stability; place of residence; type of diet; presence or absence of a community/network of support; health care; and level of education [2]. From the beginning of 2020, the COVID-19 pandemic has exacerbated and amplified social and health inequalities in various nations of the world [3,4,5,6]. Social vulnerability has proven to be a relevant risk factor for severe manifestations of COVID-19, even in children, reaffirming the notion that some SDH contribute to premature death more than the quality of healthcare received [7,8].

Due to the need to limit the spread of the SARS-CoV-2 infection, distance learning (DL) became the prevailing schooling modality during the first months of the pandemic. (Italy, from 8th March 2020, was the first country in Europe to implement a nationwide lockdown) [9] and is still utilized in cases of quarantine for groups of children in countries including Italy. DL (also known as e-learning, distance education, or online learning), is characterized by a physical separation between the students and teachers during the process of education [10], thus representing a viable alternative to the conventional lessons where presence is required. It reshaped the education service, but also provided a possible further factor of social inequity [11,12].

The aim of the present study was to investigate the experiences encountered by families with children utilizing the DL method during the months of the first wave and nationwide lockdown due to the COVID-19 pandemic in Italy, namely from March to May 2020, and to search for correlations with purported experiences and respective levels of SDH.

## 2. Materials and Methods

From 1 September to 15 October 2020, a cross-sectional online survey of 78 questions was proposed to a sample population of parents with children up to the age of 18 years and domiciled throughout the Italian national territory. The survey was developed on the online platform Qualtrics and distributed through dissemination of a link and a QR code in a tertiary care pediatric hospital and in the waiting rooms of the pediatricians’ offices, and through social networks with a snowball sampling technique. It queried the demographic variables of families, various SDH (housing, level of parental education, employment, hunger, stress), topics related to children’s behavior during lockdown (sleep, tics, changes in eating patterns), and parents’ attitude and opinions towards DL. The methods and the questionnaire have already been described in detail elsewhere [13]. Here we examined the questions about DL and measured the degree of interrelation of these questions to the available information on SDH. The parents’ level of education was measured according to the Italian school system division: secondary school normally lasts from 11 to 13 years old, while high school lasts between 14 and 19 years old.

### 2.1. Statistical Analysis

Numerical variables were summarized as mean ± standard deviation; categorical variables were summarized as counts (percentages). Comparisons between groups were performed with Fisher’s exact test, Mann–Whitney *U* test and Kruskal–Wallis test, where appropriate. Although the purpose of this analysis is mainly exploratory [14], all *p* values were corrected using the Simes–Benjamini–Hochberg method for false discovery rate control [15,16,17], which has the advantageous property that the power to detect an effect of a given size does not tend to zero as the number of comparisons increases, in contrast to the case with most other multiple–test procedures [18]. All data were analyzed using Stata version 15 (StataCorp. 2017. Stata Statistical Software: Release 15. College Station, TX, USA: StataCorp LP). The significance level was set at 5%.

### 2.2. Ethics

The present study was approved by the Ethics Committee of the IRCCS Azienda Ospedaliero-Universitaria di Bologna, Policlinico Sant’Orsola (Bologna, Italy) (Institutional Reviewer Board approval number 762/2020/Oss/AOUBo).

## 3. Results

Overall, 4031 parents of school-age children filled in the survey. After excluding incomplete answers about SDH and DL responses (country of origin, digital skills, major barriers to DL, feelings of inadequacy, and family bonds), 3791 questionnaires (94.0%) were included in the analysis.

### 3.1. Parents of Italian Origin vs. Parents from Other Countries

One hundred nineteen parents (3.1%) were not of Italian origin: 88 (73.9%) came from other European countries, 13 (10.9%) from Latin America, 6 (5.0%) from Northern America, 6 (5.0%) from Asia, and 5 (4.2%) from Africa, while 1 was of unspecified origins. A total of 88.2% of these subjects had lived in Italy for at least 5 years (*n* = 105) and 37.0% had Italian citizenship (*n* = 44).

As shown in Table 1, we observed a significant difference between the non-Italian parents who felt they did not have enough digital skills to follow their children in DL and the Italian population with the same concern (20.2% vs. 10.3%, *p* = 0.01). Similarly, non-Italian parents complained more frequently of a lack of good-quality digital equipment (12.6% vs. 5.5%, *p* = 0.01) and reported an overall lower number of personal computers (*p* = 0.04). As compared with Italians, non-Italian parents had more difficulties with the Italian language (11.8% vs. 0.1%, *p* < 0.001), and reported an insufficient level of education to support the school commitments of their children (17.6% vs. 5.7%, *p* < 0.001).

### 3.2. Parents Living Together vs. Not Living Together

Among families with parents living apart, children were reported to have greater difficulties in concentrating during DL (60.6% vs. 55.1% of the children of parents living together, *p* = 0.05) and parents referred to a lower capacity for commitment in support of DL (11.4% vs. 6.8%, *p* = 0.002) (Table 2).

### 3.3. House Property

We observed a greater propensity to declare insufficient parental digital skills, lower availability of good-quality digital devices, and greater sense of inadequacy in providing adequate digital tools in families living in houses for rent, compared to those who lived in owned property (19.9% vs. 9.5%, *p* < 0.001, 13.7% vs. 4.7%, *p* < 0.001, and 8.3% vs. 3.8%, *p* < 0.001, respectively) (Table 3).

### 3.4. Parental Education

Parental level of education (i.e., the highest level of education successfully completed according to the Italian school system) exhibited a large number of significant associations with perceived impairment in DL (Table 4). The number of computers and tablets owned by the family increased progressively with the level of education acquired by the parent (personal computer: secondary school 1.5 ± 0.8 SD, high school 1.8 ± 1.0 SD, university degree 2.2 ± 1.1 SD, *p* < 0.001).

As reported in Table 4, the barriers for children’s DL (e.g., problems in connecting to the Internet, insufficient digital facilities) increased as the parental level of education decreased; parents with lower levels of education felt more incapable of supporting their children with DL, but, on the other hand, the concomitance of DL with parental working represented a major problem for children from families with higher levels of education. In this group, parents also reported more difficulties in controlling their children’s anxiety, due to the scarcity of time spent with them.

### 3.5. Family Economic Status

The economic status of the family was assessed based on the personal perception of the family’s economic situation (well-off, average, difficult) (Table 5). Parents with a better financial situation seemed to offer more support by providing adequate digital structures to their children, both in terms of the number of devices owned by the family, and in terms of the ability to be connected to the Internet. A more prosperous family environment, in general, was associated with higher parental self-confidence in supporting children and keeping their anxieties under control. Finally, this group of parents reported better agreement with partners concerning the management of their children.

## 4. Discussion

At the beginning of the COVID-19 pandemic, the abrupt schooling transition to DL represented one of the greatest disruptions to youths’ lives [19,20], and a great challenge for parents and caregivers, who were suddenly required to support their children in DL in different ways, according to their offspring age, as well as for teachers, who were often technologically unprepared and used to didactic methods suitable for lessons in presence. This study strived to identify the effects and impact of the COVID-19 pandemic on SDH and DL in Italy. Numerous studies have analyzed the mental and emotional health of children and adolescents during the pandemic [21,22]. Other studies have focused on the students’ opinion about DL, mostly in post-secondary education [23,24,25].

After a thorough search on the PubMed database, we identified 12 studies investigating the relationship between SDH and DL at school age (primary to higher secondary grades) in different countries worldwide, although the age of schoolers and the considered parameters are highly variable. Papers evaluating this relationship are reported in Table 6. In general, a lower socioeconomic status and lower parental education are burdened by more impediments to DL in terms of lower digital devices’ availability, problems with internet connection, and parental difficulties in supporting their children’s DL. The included realities are both high-, middle-, and low-income countries; however, no information is reported about the situation in Italy.

Our study highlighted the differences in access to digital devices in relation to SDH in a wide population sample distributed all over the Italian territory. The lack of adequate equipment seems to correlate with parents being of non-Italian origin, parents not having reached a higher level of education, families not living in owned housing, and parents perceiving their economic situation as difficult. In 2018, a survey in American colleges reported that about 20% of students had issues in accessing essential technology for DL (laptops and high-speed Internet) [38]. In the same study it was shown that students of lower socioeconomic status experienced disproportionate difficulties and a correlation between malfunctioning laptops and lower grade averages. In another study based on an analysis of the Facebook posts of a population of parents, the technological barrier also appeared to be the most common limitation in supporting e-learning [39]. The first step to successful DL is reliable technology and the infrastructure to ensure the online delivery of lessons.

According to a report by the Italian National Institute of Statistics (ISTAT), relating to the period 2018–2019, 33.8% of families in Italy do not have computers or tablets at home; this rate decreases to 14.3% among families with at least a minor. Only 22.2% of families have a PC or tablet available for each family member. In Southern Italy, 41.6% of families have no computer at home (compared to an average of about 30% in other areas of the country), and only 14.1% have at least one computer for each family member [40]. In 2020, the EU Kids Online Survey mapped the internet access, online practices, skills, online risks, and opportunities for children aged 9–16 in 19 European countries (including Italy) [41]. In Italy, 84% of participants are reported to go online primarily from their smartphones. In a recent UNICEF discussion paper, 15 studies were reviewed about how children log on to the Internet. The researchers reported great difficulty in finding high-quality studies about access inequalities among children, most of all because the majority part of these reports provide only descriptive statistics, and the socioeconomic status or other SDH was hard to measure in a way that is consistent across different countries [42].

The economic state of a family is a fundamental SDH, showing that families of better economic standing were able to offer superior global digital support to their children. Although not new, this fact should further emphasize that adequate access to the web has been shown to affect the health of people and communities [43]. On the other hand, it has long been known that use of digital technology is a key driver of change and can underpin the realization of the Sustainable Development Goals promoted by the United Nations [44]. The COVID-19 pandemic has exposed how the lack of internet access could affect most of the six domains of the SDH defined by the American Medical Association [2].

In our study, several factors (migration, financial instability, house property, knowledge of the Italian language, separation of parents, level of education) included in common SDH frameworks appeared to influence the ability of parents to support their children in DL during the lockdown period. A multicenter study based on an online self-report questionnaire explored adolescents’ emotional reactions during lockdown in a sample of 2105 secondary school students (aged 14–19) in Italy, Croatia, and Romania. It highlighted how the students who believed that family support was relevant to overcome this period (73.4% of the Italian interviewees) were less likely to report both boredom/emptiness and anxiety compared to those who had not recognized such an important role of the family [45].

The findings determining DL among children from families not living together coincides with research showing lower scores and measures of academic achievement in children of single-parent households, when compared to children in two-parent households. Moreover, high rates of family fragmentation, worldwide, account for multiple socio-economic consequences, compromising economies, hindering innovation, as well as competition and growth.

Although a child’s home environment, stability, and surroundings superseded home ownership in the achievement of improved social child outcomes, it was found that subsidized home ownership led to improved outcomes, in so far as the child maintained a suitable residing environment [46]. The findings of this study, in this respect, correlate to existing work on childhood home stability. The very apparent correlations to parental levels of education and being economically viable enough to be in possession of more digital devices and being more able to assist with school exercises, are seen. Indeed, those households with higher levels of parental levels of education have ensuing resource and employment opportunities that influence cognition and social outcomes but may have significant effects on children’s behavior and receptiveness.

According to our results, in general a more prosperous family economic context seems to favor DL in several respects. This concurs with the studies that highlight strong evidence of associations between income factors and COVID-19 pandemic outcomes [47,48,49]. Furthermore, according to the latest report from ISTAT, Italy has one of the highest school dropout rates in Europe (about 13.1%) [50], which is measured by the number of 18- to 24-year-olds who have, at most, a lower secondary qualification, and are outside the training and education system (Early Leavers from Education and Training, ELET). One of the benchmarks of the Strategy EUROPE 2020 is to reach a common European target of ELET below 9% by 2031 [51]. The report does not exclude the hypothesis that the various barriers to DL may have further worsened the Italian rates of school dropout.

The present study has several limitations. This survey was spread predominantly by social media. We, therefore, do not know the true response rate. The study design could limit the ability to reach groups of the population that do not have access to the Internet and who were probably not able to follow DL in a beneficial way; indeed, the foreign-born population in Italy is estimated at around 10% by the National Institute of Statistics (accessible via web-page http://dati.istat.it/, last accessed on 24 April 2022), a value much higher than the percentage of people with an immigration background observed in our study sample (3.1%). Moreover, we did not include all parameters of SDH, but we focused on the ones that we believed to be most relevant to this pandemic. Lastly, we only obtained parental point-of-views about DL. Despite its limitations, our work confirms and analyzes the difficulties in DL and its correlation to SDH during the COVID-19 pandemic.

## 5. Conclusions

Despite the disruption in the school system brought about by the COVID-19 pandemic, it brought opportunities for innovation and resilience in education. However, unequal access to DL poses barriers to education among disadvantaged groups. From a parental point of view, during the COVID-19 pandemic, SDH seemed to have had great impacts on DL, both in terms of accessibility (digital devices availability, adequate Internet connection), and parental ability to effectively support their children. School closures affected all children, but not all in the same way: students from families with financial difficulties and low levels of parental education, or even those living in houses for rent or having separated parents, may be disadvantaged in a school setting since the introduction of DL. When remote learning cannot be avoided due to public health requirements, institutions should be aware of those groups who may experience augmented impairment of effective learning and take measures to avoid the increase in social inequities. Moreover, obstacles in DL might likely lead to disproportionate learning losses and, finally, to long lasting inequalities with consequences in adult life, reflecting both at an individual and societal level. Thus, all stakeholders, including governmental, national, and local authorities, should take action to minimize these gaps and ensure the right of education for everyone.

## Figures and Tables

**Table 1 ijerph-19-05741-t001:** Sociodemographic characteristics, parents’ feelings, and children’s distance learning (DL) experience of the study sample during the first 6 months of the COVID-19 pandemic in Italy, according to parental country of origin.

Characteristic	All (*n* = 3791)	Country of Origin		*p* Value
		Italy	Outside Italy	
		(*n* = 3672)	(*n* = 119)	
Age, y (mean ± SD)	43.2 ± 6.3	43.3 ± 6.3	40.0 ± 6.6	<0.001 **
Female sex	3471 (91.6%)	3357 (91.4%)	114 (95.8%)	0.13
Digital devices owned by the family				
Personal computers (mean ± SD)	2.0 ± 1.1	2.0 ± 1.1	1.8 ± 1.0	0.04 *
Tablets (mean ± SD)	1.4 ± 0.7	1.4 ± 0.7	1.5 ± 0.7	0.63
Lack of adequate digital skills to support children’s DL	404 (10.7%)	380 (10.3%)	24 (20.2%)	0.01 *
Major barriers to DL				
Lack of good-quality digital devices	218 (5.8%)	203 (5.5%)	15 (12.6%)	0.01 *
Lack of stable Internet connection	787 (20.8%)	761 (20.7%)	26 (21.8%)	0.77
Concentration difficulties	2112 (55.7%)	2036 (55.4%)	76 (63.9%)	0.11
Parents’ working duty and lack of children’s autonomy in DL	1672 (44.1%)	1626 (44.3%)	46 (38.7%)	0.33
Feeling of inadequacy in supporting children’s school commitments	2173 (57.3%)	2095 (57.1%)	78 (65.5%)	0.11
Reasons for the sense of inadequacy				
Incapacity in providing enough digital devices	164 (4.3%)	152 (4.1%)	12 (10.1%)	0.01 *
Lack of digital skills	230 (6.1%)	221 (6.0%)	9 (7.6%)	0.52
Insufficient educational level	232 (6.1%)	211 (5.7%)	21 (17.6%)	<0.001 **
Difficulty in the Italian language	16 (0.4%)	2 (0.1%)	14 (11.8%)	<0.001 **
Concomitance of DL with parental working	1560 (41.2%)	1523 (41.5%)	37 (31.1%)	0.05 *
Inadequacy in keeping children’s anxieties under control	1482 (39.1%)	1435 (39.1%)	47 (39.5%)	0.92
Reasons for the poor control of children’s anxiety				
Feeling of inability	227 (6.0%)	214 (5.8%)	13 (10.9%)	0.06
Fear of losing the job	172 (4.5%)	159 (4.3%)	13 (10.9%)	0.01 *
More frequent disagreement with the partner on how to behave with the children	1277 (33.7%)	1226 (33.4%)	51 (42.9%)	0.07

** *p* < 0.001; * *p* < 0.05. SD, standard deviation.

**Table 2 ijerph-19-05741-t002:** Sociodemographic characteristics, parents’ feelings, and children’s distance learning (DL) experience of the study sample during the first 6 months of the COVID-19 pandemic in Italy according to parents’ cohabitation.

Characteristic	All (*n* = 3791)	Cohabiting Parents		*p* Value
		Yes	No	
		(*n* = 3360)	(*n* = 431)	
Age, y (mean ± SD)	43.2 ± 6.3	43.1 ± 6.3	44.1 ± 6.6	<0.001 ***
Digital devices owned by the family				
Personal computers (mean ± SD)	2.0 ± 1.1	2.0 ± 1.1	1.7 ± 0.9	<0.001 ***
Tablets (mean ± SD)	1.4 ± 0.7	1.5 ± 0.7	1.4 ± 0.6	0.06
Lack of adequate digital skills to support children’s DL	404 (10.7%)	334 (9.9%)	70 (16.2%)	<0.001 ***
Major barriers to DL				
Lack of good-quality digital devices	218 (5.8%)	182 (5.4%)	36 (8.4%)	0.03 *
Concentration difficulties	2112 (55.7%)	1851 (55.1%)	261 (60.6%)	0.05 *
Parent’s working duty and lack of children’s autonomy in DL	1672 (44.1%)	1514 (45.1%)	158 (36.7%)	0.002 **
Feeling of inadequacy in supporting children’s school commitments	2173 (57.3%)	1908 (56.8%)	265 (61.5%)	0.08
No effort to support children in DL	277 (7.3%)	228 (6.8%)	49 (11.4%)	0.002 **
Inadequacy in keeping children’s anxieties under control	1482 (39.1%)	1328 (39.5%)	154 (35.7%)	0.14
Reasons for the poor control of children’s anxiety				
Fear of losing the job	172 (4.5%)	142 (4.2%)	30 (7.0%)	0.02 *
More frequent arguments with partner	1422 (37.5%)	1321 (39.3%)	101 (23.4%)	<0.001 ***
More frequent arguments with children	1431 (37.7%)	1305 (38.8%)	126 (29.2%)	<0.001 ***

*** *p* < 0.001; ** *p* < 0.01; * *p* < 0.05. SD, standard deviation.

**Table 3 ijerph-19-05741-t003:** Sociodemographic characteristics, parents’ feelings, and children’s distance learning (DL) experience of the study sample during the first 6 months of COVID-19 pandemic in Italy according to type of accommodation.

Characteristic	All (*n* = 3791)	Type of Accommodation		*p* Value
		Own House	Rented House or Similar	
		(*n* = 3369)	(*n* = 422)	
Age, y (mean ± SD)	43.2 ± 6.3	43.4 ± 6.3	41.6 ± 6.4	<0.001 **
Digital devices owned by the family				
Personal computers (mean ± SD)	2.0 ± 1.1	2.0 ± 1.1	1.7 ± 1.0	<0.001 **
Tablets (mean ± SD)	1.4 ± 0.7	1.5 ± 0.6	1.3 ± 0.6	0.007 *
Lack of adequate digital skills to support children’s DL	404 (10.7%)	320 (9.5%)	84 (19.9%)	<0.001 **
Major barriers to DL				
Lack of good-quality digital devices	218 (5.8%)	160 (4.7%)	58 (13.7%)	<0.001 **
Lack of stable Internet connection	787 (20.8%)	705 (20.9%)	82 (19.4%)	0.52
Feeling of inadequacy in supporting children’s school commitments	2173 (57.3%)	1917 (56.9%)	256 (60.7%)	0.15
Reasons for the sense of inadequacy				
Incapacity in providing enough digital devices	164 (4.3%)	129 (3.8%)	35 (8.3%)	<0.001 **
Lack of digital skills	230 (6.1%)	197 (5.8%)	33 (7.8%)	0.15
Insufficient educational level	232 (6.1%)	190 (5.6%)	42 (10.0%)	0.002 *
Difficulty with the Italian language	16 (0.4%)	9 (0.3%)	7 (1.7%)	0.002 *
Concomitance of DL with parental working	1560 (41.2%)	1420 (42.1%)	140 (33.2%)	<0.001 **
Inadequacy in keeping children’s anxieties under control	1482 (39.1%)	1300 (38.6%)	182 (43.1%)	0.09
Reasons for the poor control of children’s anxiety				
Fear of losing the job	172 (4.5%)	135 (4.0%)	37 (8.8%)	<0.001 **
More frequent disagreement with the partner on how to behave with the children	1277 (33.7%)	1106 (32.8%)	171 (40.5%)	0.003 *

** *p* < 0.001; * *p* < 0.01. SD, standard deviation.

**Table 4 ijerph-19-05741-t004:** Sociodemographic characteristics, parents’ feelings towards distance learning (DL) and children’s DL experience of the study sample during the first 6 months of the COVID-19 pandemic in Italy according to parental level of education.

Characteristic	All (*n* = 3791)	Level of Education			*p* Value
		Secondary School	High School	University	
		(*n* = 199)	(*n* = 1523)	(*n* = 2069)	
Age, y (mean ± SD)	43.2 ± 6.3	42.4 ± 7.5	43.1 ± 6.7	43.3 ± 5.9	0.29
Digital devices owned by the family					
Personal computers (mean ± SD)	2.0 ± 1.1	1.5 ± 0.8	1.8 ± 1.0	2.2 ± 1.1	<0.001 ***
Tablets (mean ± SD)	1.4 ± 0.7	1.3 ± 0.6	1.4 ± 0.7	1.5 ± 0.7	0.05 *
Lack of adequate digital skills to support children’s DL	404 (10.7%)	61 (30.7%)	220 (14.4%)	123 (5.9%)	<0.001 ***
Major barriers to DL					
Lack of good-quality digital devices	218 (5.8%)	25 (12.6%)	121 (7.9%)	72 (3.5%)	<0.001 ***
Lack of stable Internet connection	787 (20.8%)	68 (34.2%)	384 (25.2%)	335 (16.2%)	<0.001 ***
Parent’s working duty and lack of children’s autonomy in DL	1672 (44.1%)	45 (22.6%)	572 (37.6%)	1055 (51.0%)	<0.001 ***
Feeling of inadequacy in supporting children’s school commitments	2173 (57.3%)	137 (68.8%)	913 (59.9%)	1123 (54.3%)	<0.001 ***
Reasons for the sense of inadequacy					
Incapacity in providing enough digital devices	164 (4.3%)	19 (9.5%)	99 (6.5%)	46 (2.2%)	<0.001 ***
Lack of digital skills	230 (6.1%)	20 (10.1%)	124 (8.1%)	86 (4.2%)	<0.001 ***
Insufficient educational level	232 (6.1%)	59 (29.6%)	131 (8.6%)	42 (2.0%)	<0.001 ***
Difficulty with the Italian language	16 (0.4%)	4 (2.0%)	6 (0.4%)	6 (0.3%)	0.02 *
Concomitance of DL with parental working	1560 (41.2%)	48 (24.1%)	580 (38.1%)	932 (45.0%)	<0.001 ***
Inadequacy in keeping children’s anxieties under control	1482 (39.1%)	81 (40.7%)	584 (38.3%)	817 (39.5%)	0.70
Reasons for the poor control of children’s anxiety					
Spending not enough time with children	332 (8.8%)	10 (5.0%)	114 (7.5%)	208 (10.1%)	0.005 **
Feeling of inability	227 (6.0%)	17 (8.5%)	120 (7.9%)	90 (4.3%)	<0.001 ***
Fear of losing the job	172 (4.5%)	10 (5.0%)	89 (5.8%)	73 (3.5%)	0.005 **

*** *p* < 0.001; ** *p* < 0.01; * *p* < 0.05. SD, standard deviation.

**Table 5 ijerph-19-05741-t005:** Sociodemographic characteristics, parents’ feelings, and children’s distance learning (DL) experience of the study sample during the first 6 months of the COVID-19 pandemic in Italy according to household economic status.

Characteristic	All (*n* = 3791)	Household Economic Status			*p* Value
		Well-off	Average	Difficult	
		(*n* = 1421)	(*n* = 2075)	(*n* = 295)	
Age, y (mean ± SD)	43.2 ± 6.3	43.8 ± 6.1	42.9 ± 6.3	42.4 ± 7.3	<0.001 **
Digital devices owned by the family					
Personal computers (mean ± SD)	2.0 ± 1.1	2.3 ± 1.2	1.9 ± 1.0	1.6 ± 0.9	<0.001 **
Tablets (mean ± SD)	1.4 ± 0.7	1.5 ± 0.7	1.4 ± 0.7	1.3 ± 0.6	<0.001 **
Lack of adequate digital skills to support children’s DL	404 (10.7%)	59 (4.2%)	251 (12.1%)	94 (31.9%)	<0.001 **
Major barriers to DL					
Lack of adequate digital skills to support children’s DL	218 (5.8%)	34 (2.4%)	120 (5.8%)	64 (21.7%)	<0.001 **
Lack of stable Internet connection	787 (20.8%)	212 (14.9%)	468 (22.6%)	107 (36.3%)	<0.001 **
Concentration difficulties	2112 (55.7%)	738 (51.9%)	1199 (57.8%)	175 (59.3%)	0.001 *
Feeling of inadequacy in supporting children’s school commitments	2173 (57.3%)	730 (51.4%)	1229 (59.2%)	214 (72.5%)	<0.001 **
Reasons for the sense of inadequacy					
Incapacity in providing enough digital devices	164 (4.3%)	19 (1.3%)	101 (4.9%)	44 (14.9%)	<0.001 **
Lack of digital skills	230 (6.1%)	60 (4.2%)	135 (6.5%)	35 (11.9%)	<0.001 **
Insufficient educational level	232 (6.1%)	39 (2.7%)	158 (7.6%)	35 (11.9%)	<0.001 **
Difficulty with the Italian language	16 (0.4%)	2 (0.1%)	8 (0.4%)	6 (2.0%)	<0.001 **
Inadequacy in keeping children’s anxieties under control	1482 (39.1%)	495 (34.8%)	828 (39.9%)	159 (53.9%)	<0.001 **
Reasons for the poor control of children’s anxiety					
Feeling anxious	816 (21.5%)	264 (18.6%)	462 (22.3%)	90 (30.5%)	<0.001 **
Spending not enough time with children	332 (8.8%)	124 (8.7%)	174 (8.4%)	34 (11.5%)	0.21
Feeling of inability	227 (6.0%)	67 (4.7%)	128 (6.2%)	32 (10.8%)	<0.001 **
Uncertainties about the future	590 (15.6%)	166 (11.7%)	345 (16.6%)	79 (26.8%)	<0.001 **
Fear of losing the job	172 (4.5%)	33 (2.3%)	104 (5.0%)	35 (11.9%)	<0.001 **
Fear of being infected with COVID-19	382 (10.1%)	135 (9.5%)	214 (10.3%)	33 (11.2%)	0.57
Fear of loved ones being infected with COVID-19	595 (15.7%)	194 (13.7%)	336 (16.2%)	65 (22.0%)	0.001 *
More frequent disagreement with the partner on how to behave with the children	1277 (33.7%)	447 (31.5%)	708 (34.1%)	122 (41.4%)	0.005 *

** *p* < 0.001; * *p* < 0.01. SD, standard deviation; COVID-19, coronavirus disease 2019.

**Table 6 ijerph-19-05741-t006:** Studies investigating the relationship between social determinants of health (SDH) and distance learning (DL) during the COVID-19 pandemic.

Study, Year	Study Type	School Grade/Children’s Age	N	Country	Included SDH	Main Results
Ariyo, E., 2022 [26]	Cross-sectional online survey	Primary and secondary school	1121 parents/caregivers	Nigeria	Parental age and educational status, household socioeconomic status, household geographical location and household size	Household size and perceived socioeconomic status of parents were related to engagement in DL; household wealth was associated with all types of activity engagement.
Azubuike, O.B., 2021 [27]	Cross-sectional survey online + telephone	Primary (25%), junior secondary (14%), higher secondary (26%), and higher (38%) schools–mean age 16.8 years	557 students and 626 parents	Nigeria	Parental educational level, region of residence, public vs. private school (proxy for socioeconomic status)	Relationship between socioeconomic status and digital divide in accessing remote learning (affordability of phone credit and internet data, electricity, access to devices); association between parental level of education and the ability to support children’s DL.
Bonal, X., 2020 [28]	Cross-sectional online survey	3–18 years old	35,419 parents/caregivers	Spain	Home size, availability of outdoor spaces, access to the Internet and to digital devices	Middle-class families were able to maintain higher standards of education quality, while children from socially disadvantaged families had few learning opportunities
Busko, V., 2021 [29]	Cross-sectional online survey	High school	4492 teens	Croatia	Availability of digital technology, internet access, housing conditions	Limited availability of digital devices, problems with internet access, missing a quiet room contributed to stress generated by DL
Haelermans, C., 2022 [30]	Cross-sectional	Primary school	201,819 students undergoing standardized tests in reading, spelling, and math	Netherlands	Migration background, parental income, parental education, urbanization	Children from lower-educated and poorer families had less access to additional resources at home
Jones, N., 2021 [31]	Longitudinal mixed-methods study via phone interviews	13–20 years	3066 adolescents	Ethiopia	Urban vs. rural residence, return to school at re-opening, sex, disability	Rural adolescents, girls, and those with disabilities were less likely to have access to DL (connectivity challenges, discriminatory gender norms, lack of adaptation for disability), and to return to school at reopening
Ma, Z., 2021 [32]	Cross-sectional online survey	7–15 years	668 parents/caregivers	China	Ethnicity, parents’ marital status, level of education, profession, and income	Residential background and family income are significantly linked with effectiveness and satisfaction of DL
Morse, A.R., 2022 [33]	Longitudinal online survey	Primary or high school	176 parents/caregivers	Australia	Housing, parental employment, stress, internet access, level of education	Increased stress associated with the difficulty of managing work-life-school balance, not having enough time to do everything, and juggling the varying needs of multiple children; technological limitations.
Poulain, T., 2021 [34]	Longitudinal online survey	1–10 years old	285 parents/caregivers	Germany	Socioeconomic status (combination of information about parental education, occupational status, and household income)	Lower socioeconomic status associated with reduced time doing schoolwork and reduced ability to concentrate on schoolwork
Sanrey, C., 2021 [35]	Cross-sectional online survey	Preschool to elementary school	360 parents/caregivers	France	Digital equipment, social position index	Lower social position associated with more time spent homeschooling children, lower digital equipment, and feeling less capable of homeschooling. Higher social position associated with children spending more time doing activities considered to be “educationally profitable”, and less time doing “unprofitable activities”.
Smetackova, I., 2021 [36]	Cross-sectional online survey	Primary school	2528 parents/caregivers	Czech Republic	Parental education and occupation, household composition, housing, devices availability, internet connection, socioeconomic status	Concerns about children’s school results were expressed by parents with low more than by those with high socioeconomic status.
Vogelbacher, 2022 [37]	Longitudinal study (computed-assisted interviews)	Second grade	1812 families	Germany	Family socioeconomic status, level of education, home learning environment, preceding parental stress	Higher educated parents and parents with lower socioeconomic status reported more stress during school closure; parental preceding stress and higher level of education associated with higher ability to support children during DL

## Data Availability

The data presented in this study are available on request from the corresponding author. The data are not publicly available due to the accordance with the Ethics Committee policy.

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
