# Peer review of "Social Determinants of Health and Distance Learning in Italy in the Era of the SARS-CoV-2 Pandemic"

_ijerph, 2022, doi:10.3390/ijerph19095741_

Round 1

Reviewer 1 Report

Dear authors,

Your manuscript addresses a thematic field, which has been reasonably in the focus of global academic and political concern in the last two years, yet empirical research is still needed in order to better understand the real-life efficacy of dynamically changing measures in the area of health and education. Your paper offers a significant contribution on the social determinants of health and distance learning, providing as a case study the situation in Italy during pandemic. 

However, I would encourage you to expand the introduction and to write a section of literature review in order to better present the frame of the already existing academic outputs in the field you study. Also, you may take into consideration conducting a comapartive analysis with some other countries.

I also suggest you extend the Conclusion section in order to better present the implications of your study for the society and not only.

Regards,

Author Response

- Your manuscript addresses a thematic field, which has been reasonably in the focus of global academic and political concern in the last two years, yet empirical research is still needed in order to better understand the real-life efficacy of dynamically changing measures in the area of health and education. Your paper offers a significant contribution on the social determinants of health and distance learning, providing as a case study the situation in Italy during pandemic.

Author response: Thank you for your appreciation and valuable comments.

- However, I would encourage you to expand the introduction and to write a section of literature review in order to better present the frame of the already existing academic outputs in the field you study. Also, you may take into consideration conducting a comapartive analysis with some other countries.

Author response: Thank you for your suggestion, which gave us the opportunity to broaden the paper. We performed a structured search on Pubmed and retrieved 12 papers discussing about SDH and DL. We added this point and a detailed table (including the country where the study was based) at the beginning of the discussion. The table synthesizes the main relationships found by the authors between the evaluated SDH and barriers to DL.

- I also suggest you extend the Conclusion section in order to better present the implications of your study for the society and not only.

Author response: Following your suggestion, we expanded the conclusion section and discussed about the possible implications of difficulties in DL.

Reviewer 2 Report

I want to thank the authors and editors for their work and the opportunity to review this manuscript. 
After a short summary of the article, I'll first address major points before listing minor issues.

================================================================================
Summary and overall assessment
================================================================================
The current research article reports associations between selected social determinants of health (SDH) and parents' experience with their children's forced switch to distance learning (DL) during the first nation-wide lockdown due to the SARS-CoV-2 pandemic in Italy (March to May 2020). Data was collected in Autumn 2020 with an online questionnaire resulting in 7958 responses, out of which 3791 were analyzed for this article. DL factors that went into the analyses included, e.g., digital skills and availability/number of digital devices needed for DL. SDH included in the analyses were parents' level of education, their country of origin (Italy vs. others), cohabitation status, the families' type of accomodation (rented vs. owned) and economic status.

Main findings include associations between adequate digital skills and parents' origin, cohabitation status, level of education, economic status, and type of accomodation. Similar associations were found for availability of digital devices for DL. The authors concluded that educationally and/or economically disadvantaged families also had more difficulties in adjusting to the DL setting, which would amplify their children's disadvantage even further.

---

I find the article a valuable contribution in the investigation into factors regarding how the pandemic-related sudden forced switch from presence teaching to DL in school children affected families and children. My main concerns are with the statistics and reporting of the results, but I think they can be adequately addressed in a revision of the manuscript. 

================================================================================
MAJOR COMMENTS
================================================================================

1) Multiple comparisons problem:
The reporting lacks any mention of the multiple comparisons/testing problem and there was no correction applied for it. E.g., for Table 1 the authors ran (excluding the sociodemographic variables) 17 individual significance tests between two groups. With a Bonferroni-corrected alpha level for 17 tests (0.05/17 = 0.0029), one would need p-values < 0.0029 to consider an effect significant. Applying this correction for the tests in Table 1, only the associations with digital skills (p = 0.002) and perceived inadequacy due to education level and language difficulties (both p < 0.001) would still be considered statistically significant.
I only used the Bonferroni-correction procedure above to illustrate the issue. Applying a more moderate procedure like a correction based on the false discovery rate (FDR) might be more appropriate in this case. See, for example, Cao & Zhang (2014), Streiner & Norman (2011), and Bender & Lange (2001) for general discussions of the problem and comparisons of procedures to counter it, and Benjamini & Hochberg (1995) for a common procedure to control the FDR. 

While it is debatable whether using a correction procedure is needed when doing purely exploratory data analyses, I think the article would benefit from at least discussing this problem, especially considering that a finding with p = 0.04 is presented prominently in the abstract. One can argue, in line with the conclusion by Streiner & Norman (2011): "If a small number of hypotheses have been stated a priori or if the purpose of the study is exploratory, then such corrections are probably not needed." But in that case, it should be stated more clearly that the whole study was purely exploratory and that any interpretation of these findings should be treated very carefully.

In a previous publication by the same research group that used data from the same questionnaire, the authors wrote: "To control for type I error related to multiple testing, the significance level was set at 0.01."(Dondi et al. (2021), p.3) I would personally recommend discussing the multiple comparison problem somewhere in the article. Then, I would apply the Benjamini-Hochberg procedure for all tests involving the same research question, which here is given for all tests reported in the same Table (probably excluding the sociodemographic variables). Choosing a reasonable FDR one would then know which results can still be considered statistically significant, keeping false positives at bay.

2) I wondered how representative the involved sample and subsamples were. The authors already mention a limitation given how participants for the survey were recruited via social media. Given that the overall sample is fairly large, it might still be reasonably representative besides such limitations. But I wondered how it is with subsamples that were used in individual analyses. E.g., in Table 1 we have the group with non-italian origin (N=111). How representative is that subsample (3.1% of whole sample) for the same subsample in the whole Italian population? Such considerations for this and other subsample might be expanded in the discussion section.

================================================================================
MINOR COMMENTS
Page and line numbers of the specific manuscript position are given in 
parentheses before comments as follows: page 1 lines 23 to 25 [P1L23-25]
================================================================================

INTRODUCTION
--------------------------------------------------------------------------------

3) [P2L45] In my understanding the term "home schooling" goes beyond online/distance/remote teaching and was used long before such teaching forms existed. The authors might want to exclude or clarify this term here, especially considering that they later define DL as "characterized by a physical separation between students and teachers." In home schooling the teacher is usually a parent and not usually physically separated while teaching.

METHODS
--------------------------------------------------------------------------------

4) [P2L74] "Mann-Whitney test" is usually called the "Mann-Whitney U test"

Results
--------------------------------------------------------------------------------
5) [P2L84] Please consider clarifying when the data was excluded and for which analyses. Was all data from a parent excluded from all analyses as soon as there was missing data in any of the SDH? I assume that was done here, as the N for all analyses was given as constant. Another approach would be to use the maximal number of available data for each analysis, by only excluding analysis-wise the parents with missing data for the relevant SDH. If there are large differences between the samples for the individual analyses in that case, I would consider redoing the analyses for these larger samples.

6) [P3L100-102] In my reading of the Table 1, this statement is wrong. The "main reason given for the sense of inadequacy felt by non-Italian parents" would be "Concomitance of DL with parental working" since 31.1% reported this reason and only 17.6% and 5.7% reported the other two reasons mentioned there.

7) [P4L128] Order of reporting in text doesn't match order of reporting in Table 3. 

8) [P5L144] What is the statement "SDH mostly correlated" based on? Where do we find an effect size comparison that would allow such a statement about the size/strength of the different correlations, to consider this the strongest one? 

9) [P5L171-172] "seemed to encourage parental self-confidence in supporting their children and keeping their anxieties under control, thanks to a lessened fear of losing their job". This statement assumes causality, which we can't do with correlational data and results. Also, any interpretation of the results should be left to the discussion section.

DISCUSSION & CONCLUSION
--------------------------------------------------------------------------------
10) [P6L195] "suboptimal" The results Table used the term "difficult". For readers' convenience, I recommend using the same term in the text.

11) [P6L206] "decreased up to 14.3%" - I assume there is something wrong with that.

12) [P6L209] "for each component" - Should probably be "for each family member"

13) [P7213] I fail to see how this statement about "navigation skills are low" relates to the current study and what is meant with "navigating skills" in this context in the first place. Please clarify or remove.

14) [P7L263] "target of ELET below the 9%" - "the" can be removed

References
--------------------------------------------------------------------------------
- Dondi, A., Candela, E., Morigi, F., Lenzi, J., Pierantoni, L., & Lanari, M. (2020). Parents’ perception of food insecurity and of its effects on their children in Italy six months after the COVID-19 pandemic outbreak. Nutrients, 13(1), 121. https://doi.org/10.3390/nu13010121
- Cao, J., & Zhang, S. (2014). Multiple comparison procedures. Jama, 312(5), 543-544. https://doi:10.1001/jama.2014.9440
- Streiner, D. L., & Norman, G. R. (2011). Correction for multiple testing: is there a resolution?. Chest, 140(1), 16-18.https://doi.org/10.1378/chest.11-0523
- Bender, R., & Lange, S. (2001). Adjusting for multiple testing—when and how?. Journal of clinical epidemiology, 54(4), 343-349. https://doi.org/10.1016/S0895-4356(00)00314-0
- Y. Benjamini and Y. Hochberg. “Controlling the false discovery rate: a practical and powerful approach to multiple testing.” Journal of the Royal Statistical Society Series B 57, no. 1 (1995): 289–300. http://www.jstor.org/stable/2346101 https://doi.org/10.1111/j.2517-6161.1995.tb02031.x

Author Response

Major Comments

1) Multiple comparisons problem. The reporting lacks any mention of the multiple comparisons/testing problem and there was no correction applied for it. E.g., for Table 1 the authors ran (excluding the sociodemographic variables) 17 individual significance tests between two groups. With a Bonferroni-corrected alpha level for 17 tests (0.05/17 = 0.0029), one would need p-values < 0.0029 to consider an effect significant. Applying this correction for the tests in Table 1, only the associations with digital skills (p = 0.002) and perceived inadequacy due to education level and language difficulties (both p < 0.001) would still be considered statistically significant. I only used the Bonferroni-correction procedure above to illustrate the issue. Applying a more moderate procedure like a correction based on the false discovery rate (FDR) might be more appropriate in this case. See, for example, Cao & Zhang (2014), Streiner & Norman (2011), and Bender & Lange (2001) for general discussions of the problem and comparisons of procedures to counter it, and Benjamini & Hochberg (1995) for a common procedure to control the FDR. While it is debatable whether using a correction procedure is needed when doing purely exploratory data analyses, I think the article would benefit from at least discussing this problem, especially considering that a finding with p = 0.04 is presented prominently in the abstract. One can argue, in line with the conclusion by Streiner & Norman (2011): "If a small number of hypotheses have been stated a priori or if the purpose of the study is exploratory, then such corrections are probably not needed." But in that case, it should be stated more clearly that the whole study was purely exploratory and that any interpretation of these findings should be treated very carefully. In a previous publication by the same research group that used data from the same questionnaire, the authors wrote: "To control for type I error related to multiple testing, the significance level was set at 0.01."(Dondi et al. (2021), p.3) I would personally recommend discussing the multiple comparison problem somewhere in the article. Then, I would apply the Benjamini-Hochberg procedure for all tests involving the same research question, which here is given for all tests reported in the same Table (probably excluding the sociodemographic variables). Choosing a reasonable FDR one would then know which results can still be considered statistically significant, keeping false positives at bay.

Author response: Thank you very much for pointing this out. We have rerun all analyses applying the Simes–Benjamini–Hochberg method for false discovery rate control, as suggested. You will notice that results are virtually coincident with those obtained in the previous version of the paper, with the exception of two variables in Table 1 that have now lost their statistical significance (feeling of inability and disagreement with the partner). Please note that difficulty in concentration is confirmed to occur differently according to parents’ cohabitation (P = 0.04502), as still reported in the abstract. Multiple-testing correction has been now addressed and discussed in the Statistical Analysis subsection as follows:

“Although the purpose of this analysis is mainly exploratory (Streiner & Norman, 2011), all P values were corrected using the Simes–Benjamini–Hochberg method for false discovery rate control (Simes, 1986; Benjamini & Hochberg, 1995), which has the advantageous property that the power to detect an effect of a given size does not tend to zero as the number of comparisons increases, in contrast to the case with most other multiple–test procedures (Genovese & Wasserman, 2002).

2) I wondered how representative the involved sample and subsamples were. The authors already mention a limitation given how participants for the survey were recruited via social media. Given that the overall sample is fairly large, it might still be reasonably representative besides such limitations. But I wondered how it is with subsamples that were used in individual analyses. E.g., in Table 1 we have the group with non-italian origin (N=111). How representative is that subsample (3.1% of whole sample) for the same subsample in the whole Italian population? Such considerations for this and other subsample might be expanded in the discussion section.

Author response: As stated in our response to comment #5, the target sample of this analysis is actually constituted by the parents of school-age children (n = 4031), a size much lower than the overall number of people who filled in the online survey (n = 7958). This inaccuracy has been now corrected in the manuscript. This implies that listwise deletion of 6.0% of records with missing data could not undermine the representativeness of our sample. However, as also stated among the study limitations, our survey could not reach groups of the population that do not have access to the internet and who are likely not able to follow DL in a beneficial way for a number of reasons (low education, language issues, etc.). We cannot know the response rate to our survey, but expanded the Limitations paragraph by mentioning that the foreign-born population in Italy is estimated at 10–11%, a value substantially higher than that observed in our study sample (3.1%).

Minor Comments

3) [P2L45] In my understanding the term "home schooling" goes beyond online/distance/remote teaching and was used long before such teaching forms existed. The authors might want to exclude or clarify this term here, especially considering that they later define DL as "characterized by a physical separation between students and teachers." In home schooling the teacher is usually a parent and not usually physically separated while teaching.

Author response: Thank you for spotting this inaccuracy. Home schooling is actually a term that indicates parents or an educator teaching at home to children whose family chose this way of schooling. This is legally allowed in several countries but is different from digital learning and out of the scope of our study. We amended the text where necessary. We are sorry for the mistake.

4) [P2L74] "Mann-Whitney test" is usually called the "Mann-Whitney U test"

Author response: The text has been amended, as suggested.

5) [P2L84] Please consider clarifying when the data was excluded and for which analyses. Was all data from a parent excluded from all analyses as soon as there was missing data in any of the SDH? I assume that was done here, as the N for all analyses was given as constant. Another approach would be to use the maximal number of available data for each analysis, by only excluding analysis-wise the parents with missing data for the relevant SDH. If there are large differences between the samples for the individual analyses in that case, I would consider redoing the analyses for these larger samples.

Author response: Thank you for this suggestion. We have now clarified at the beginning of the Results section that the target sample for this analysis was constituted by the parents of school-age children (n = 4031), a size much lower than the overall number of people who filled in the online survey (n = 7958). As now stated in the manuscript, we excluded from all analyses the questionnaires that had some missing information about country of origin, digital skills, major barriers to distance learning, feelings of inadequacy, and family bonds, that is, 240 out of 4031 (6.0%), leading to a final sample size of 3791. Although we appreciate your suggestion of performing a pairwise deletion of missing values, we think that listwise deletion is a better option when the proportion of mission data is contained and many tests or regression models are performed. Indeed, a pairwise deletion would improve the study power by just a few decimal points and provide a potential ambiguous definition of the sample size.

6) [P3L100-102] In my reading of the Table 1, this statement is wrong. The "main reason given for the sense of inadequacy felt by non-Italian parents" would be "Concomitance of DL with parental working" since 31.1% reported this reason and only 17.6% and 5.7% reported the other two reasons mentioned there.

Author response: Thank you for noticing this inaccuracy. The sentence has been rewritten as follows: “As compared with Italians, non-Italian parents had more difficulties with the Italian language (11.8% vs 0.1%, P <0.001) and reported an insufficient level of education to support the school commitments of their children (17.6% vs 5.7%, P <0.001).”

7) [P4L128] Order of reporting in text doesn't match order of reporting in Table 3. 

Author response: We have corrected the text, as suggested.

8) [P5L144] What is the statement "SDH mostly correlated" based on? Where do we find an effect size comparison that would allow such a statement about the size/strength of the different correlations, to consider this the strongest one? 

Author response: We agree with you that this is an overinterpretation of data, since we do not present directly any odds ratios. We have therefore rewritten the sentence.

9) [P5L171-172] "seemed to encourage parental self-confidence in supporting their children and keeping their anxieties under control, thanks to a lessened fear of losing their job". This statement assumes causality, which we can't do with correlational data and results. Also, any interpretation of the results should be left to the discussion section.

Author response: We agree with you that our study does not allow any causal statement. The sentence has been therefore reframed and shortened.

10) [P6L195] "suboptimal" The results Table used the term "difficult". For readers' convenience, I recommend using the same term in the text.

Author response: We amended the text using the word “difficult” in order to make it clearer to the reader.

11) [P6L206] "decreased up to 14.3%" - I assume there is something wrong with that.

Author response: This sentence was actually difficult to understand. In order to make it clearer, we changed it as follows: “…33.8% of families in Italy do not have computers or tablets at home; this rate decreases to 14.3% among families with at least a minor..

12) [P6L209] "for each component" - Should probably be "for each family member"

Author response: The text was amended according to your suggestion.

13) [P7213] I fail to see how this statement about "navigation skills are low" relates to the current study and what is meant with "navigating skills" in this context in the first place. Please clarify or remove.

Author response: Thank you for your comment. This topic is actually out of the scope of this paper and we decided to remove the sentence.

14) [P7L263] "target of ELET below the 9%" - "the" can be removed

Author response: Thank you, “the” was removed.

Round 2

Reviewer 2 Report

Major Comments
===============

1) Thanks for considering my comment and adding adjusted p-values as well as explaining this in the Statistical Analysis subsection. You now replaced all p-values with FDR-adjusted p-values based on the B-H-procedure. Since the original publication by Benjamini & Hochberg (1995) doesn't include the calculation of adjusted p-values, I suggest adding a publication that does:
- Yekutieli, D., & Benjamini, Y. (1999). Resampling-based false discovery rate controlling multiple test procedures for correlated test statistics. Journal of Statistical Planning and Inference, 82(1-2), 171-196. https://doi.org/10.1016/S0378-3758(99)00041-5
- Benjamini, Y., Heller, R., & Yekutieli, D. (2009). Selective inference in complex research. Philosophical Transactions of the Royal Society A: Mathematical, Physical and Engineering Sciences, 367(1906), 4255-4271. https://doi.org/10.1098/rsta.2009.0127

Reply: thank you for appreciating our work. We changed the reference according to your suggestion and added these two papers (Yekutieli et al and Benjamini et al).

2) Thanks for adding this issue in the Discussion section. 

Reply: thank you for helping us improving our paper.

Minor Comments
===============

All minor comments were handled carefully. Thanks for considering my suggestions and editing the manuscript accordingly where deemed appropriate, or explaining why specific suggestions were disregarded (e.g. in the case of missing values handling).

Reply: again, thank you for your precious suggestions.